# A Review of the Preparation, Characterization, and Applications of Chitosan Nanoparticles in Nanomedicine

**DOI:** 10.3390/nano13081302

**Published:** 2023-04-07

**Authors:** Rejeena Jha, Robert A. Mayanovic

**Affiliations:** Department of Physics, Astronomy & Materials Science, Missouri State University, Springfield, MO 65897, USA; rj23s@login.missouristate.edu

**Keywords:** chitosan, chitin, chitosan nanoparticle, drug delivery, cancer therapy, tissue engineering

## Abstract

Chitosan is a fibrous compound derived from chitin, which is the second most abundant natural polysaccharide and is produced by crustaceans, including crabs, shrimps, and lobsters. Chitosan has all of the important medicinal properties, including biocompatibility, biodegradability, and hydrophilicity, and it is relatively nontoxic and cationic in nature. Chitosan nanoparticles are particularly useful due to their small size, providing a large surface-to-volume ratio, and physicochemical properties that may differ from that of their bulk counterparts; thus, chitosan nanoparticles (CNPs) are widely used in biomedical applications and, particularly, as contrast agents for medical imaging and as vehicles for drug and gene delivery into tumors. Because CNPs are formed from a natural biopolymer, they can readily be functionalized with drugs, RNA, DNA, and other molecules to target a desired result in vivo. Furthermore, chitosan is approved by the United States Food and Drug Administration as being Generally Recognized as Safe (GRAS). This paper reviews the structural characteristics and various synthesis methods used to produce chitosan nanoparticles and nanostructures, such as ionic gelation, microemulsion, polyelectrolyte complexing, emulsification solvent diffusion, and the reverse micellar method. Various characterization techniques and analyses are also discussed. In addition, we review drug delivery applications of chitosan nanoparticles, including for ocular, oral, pulmonary, nasal, and vaginal methodologies, and applications in cancer therapy and tissue engineering.

## 1. Introduction

Significant developments have been made in the recent past in medical imaging, drug delivery systems, advanced therapy, and the treatment of fatal diseases. In particular, the discovery of nanomaterials and nanomedicine has dramatically improved the precision and efficacy of a significant number of medical procedures and treatments. Nanoparticles (NPs) are nano-sized (on a scale of ~10^−9^ m) particles of matter that may have very unusual mechanical, physical, optical, and chemical properties compared to their larger-sized or bulk counterparts. Due to our increased capability of tuning desired characteristics and properties at various sizes and dimensions, the uses of nanomaterials have widened across a wide extent of industrial applications, including for medicine, cosmetics, air purification, agriculture, and environmental remediation. In recent years, the application of nanoparticles has increased significantly to the point of creating a nascent field of medicine, generally termed nanomedicine. Nanoparticles are widely used as contrast agents for medical imaging applications and as transport agents for drug and gene delivery in vivo [1]. Due to their small size, nanoparticles can enter the body and reach the specific tissue more efficiently and in a more direct fashion. Nanoparticles also have the capability to deliver molecules, such as from drugs, in less time and with lower pain to detect and cure diseases [2]. For nanomaterials to be used in the treatment of disease, many factors need to be considered, such as biodegradability, biocompatibility, size, hydrophilicity, and conjugation power with various drugs. Among the water-soluble materials that are currently available, chitosan has all of the properties mentioned above. Furthermore, in part due to being inexpensive, chitosan nanoparticles are widely investigated for their implementation as drug delivery systems to cure various fatal diseases.

In this review, we discuss the synthesis, characterization, drug conjugation, and in vivo treatment of various diseases using chitosan nanoparticles. Chitosan is a polysaccharide or fibrous compound prepared by the N-deacetylation of chitin. Chitin is a biopolymer naturally produced within crustaceans’ shells, such as those of shrimps, lobsters, and crabs. Chitosan also occurs in microorganisms, such as fungi and yeast [1]. The molecular structure of chitosan consists of glucosamine and N-acetyl-glucosamine units: The repeatability of the units is determined by the degree of deacetylation (DD) (see Figure 1). Having an equilibrium acid association constant pK_a_ value of ~6.5 on the amine groups makes chitosan insoluble at neutral pH values; however, chitosan is soluble at acidic pH values < 6.5, whereas the chitosan molecule is positively charged [3]. Notably, compounds in their un-ionized form tend to be less soluble, but can more easily penetrate lipophilic barriers between them, and are a biological target of interest [4]. The degree of deacetylation directly affects the occurrence of amine groups in the chitosan molecular structure, which can be protonated [3]. This has a direct bearing on the solubility, the degree of hydrophilicity vs. hydrophobicity, and the nature of interactions of chitosan with various polyanions. Chitosan is soluble in acetic, formic, citric, tartaric, and other organic acids [5]. Chitosan, however, is insoluble in some inorganic acids, including phosphoric and sulfuric acids [6]. Chitosan can be obtained in a broad range of molecular weights and degrees of deacetylation [1]. For considerations of the synthesis of chitosan nanoparticles, molecular weight and the degree of deacetylation have a direct bearing on their particle size, the nature of particle formation, and their degree aggregation in solution [1]. 

Chitosan nanoparticles (CNPs) can be produced with the desirable nano-scale characteristics, such as a small size, certain surface and interface effects, and quantum size effects [7]. Owing to the enormous potential of CNPs in biomedical applications, including for drug delivery, gene delivery, the treatment of various diseases such as cancer therapy, and tissue engineering, this review covers the various synthesis methods and the characterization of structural and related properties of CNPs. In addition, we review the applications of CNPs in the field of nanomedicine.

## 2. Structural and Physiochemical Characteristics of Chitosan

The structural and physicochemical properties of chitosan have been investigated in detail by various researchers. The discovery of chitin was first made by Henri Braconnot in 1811 during his investigations on mushrooms [8]. In 1859, Charles Rouget discovered that the alkali treatment of chitin could form a very different organic polysaccharide that can be dissolved in acids [7]. This organic polysaccharide was termed chitosan by Hoppe Seiler [9].

Chitin is among the (i.e., second) most abundant polysaccharides in nature. It is present in the crustacean shells of shrimps, lobsters, and crabs [10]. It is also the primary constituent of insects’ cuticles, fungal cell walls, yeasts, and green algae [11]. Unlike chitin, chitosan has a much lower occurrence in nature; however, it has been discovered to occur within the cell wall of certain types of fungi [12]. Chitin is a polymer that comprises [β-(1-4)-2-acetamido-2deoxy-D-glucopyranose] units. The idealized structure of chitin is similar to that of cellulose with the exception that an acetamido group substitutes for the C(2) hydroxyl group [13]. There are three types of chitin: α, β, and γ [7]. Specifically, α-chitin has an antiparallel chain, β-chitin has intrasheet hydrogen bonding within parallel chains, and γ-chitin is the combination of both α- and β- chitin [14]. Chitosan is predominantly a derivative from chitin [13].

Chitosan contains 60% or more glucosamine (D units) [15]. The D unit content (and the free amine groups) in chitosan enables its solubility in aqueous acidic solution. The degree of deacetylation (DD) value is a reflection of the fractional molar content of D units in chitosan [16]. Thus, the DD value has a direct bearing on the performance of chitosan in a wide variety of applications [17]. The DD value of chitosan can be determined by using infrared radiation (IR) spectroscopy [18], UV-visible spectrophotometry [19], potentiometric titration [20], H-liquid-state nuclear magnetic resonance (NMR), and solid-state NMR spectroscopy [21]. Chitin is a comparatively intractable polymer that has sufficient structural dissimilarity to cellulose so that it is insoluble in solvents used to dissolve cellulose, including cuprammonium hydroxide (Scheweizer’s reagent), cadoxe, and cupriethylene diamine [22].

The molecular structures of chitin and chitosan are shown in Figure 1. The solubility of chitosan depends on various factors such as DD, pH, temperature, polymer crystallinity, and the type of solvent. Chitosan solubility in aqueous media is determined by the extent of protonated NH_2_ groups: for example, chitosan is soluble in aqueous solutions when ~50% of the protonation of amino groups occurs [23]. If DD is ~28%, chitosan is soluble in an acetic acid solution. Thus, for all other factors remaining the same, chitosan solubility is directly impacted to the degree of deacetylation since this determines the extent of glucosamine units and modifies its crystal structure [23].

The molecular weight of chitosan and its viscosity in aqueous media also have a determinative effect in the biochemical, nanomedicinal, and pharmacological applications of the polymer. Additional determinative factors include the degree of crystallinity, crystal size, ash content, moisture content, and the presence of heavy metals [24]. An additional benefit of its industrial use is that chitosan harvesting leads to ameliorating the pollution of the environment caused by the disposal of crustacians’ shells by the seafood industry [7]. Every year millions of tons of crustacean shells are produced as waste, which can degrade slowly and pollute the environment. The conversion of these shells into chitin and chitosan is one of the best solutions to combat this pollution, as chitin and chitosan have many applications in a variety of fields.

## 3. Chemistry of Chitosan

Chitosan has three reactive or functional groups: an amino group (-NH_2_) situated at C_2_-NH_2_, and two hydroxyl groups located at C_3_-OH, and C_6_-OH (Figure 1). The C_6_-OH hydroxyl group is more chemically active than the one at C_3_-OH. The glycosidic bond can also be considered a functional group that allows for chemical modifications, producing a polymer with new properties and behavior [23]. Through the use of suitable reagents, chemical modification at the amino group results in N-modified chitosan derivatives, whereas chemical modification at the hydroxyl groups results in O-modified chitosan derivatives, thus providing improved physicochemical properties [25]. Although the chemical modification of chitosan may occur at all of its functional groups, it is the reactions of side groups at the hydroxyl group sites that result in minimal or no change in its biophysical properties [7].

Chitosan functionalized by the N-cinnamyl substituting O-amine group is substantially more hydrophobic and is a substantial antimicrobial agent against *Staphylococcus aureus*, *Bacillus cereus*, *Escherichia coli*, and *Pseudomonas aeruginosa* [26]. Chitosan-based glycopolymer modifying C_6_ N-quaternary ammonium-O-sulfobetaine is soluble in aqueous media and has a good affinity for binding with lectins [27]. In addition, this modified chitosan has proven to poses antimicrobial activity [28]. O-acylated chitosan nanofibers with fatty acids anhydrate side groups have been shown to poses varying degrees of hydrophobic and hydrophilic values that directly correlate with the chain length of the substituted acyl group [29]. Chitosan quaternary ammonium salt can act as a coagulant and flocculant agent that is effective against *Microcystis aeruginosa* cyanobacteria [30]. The quaternary ammonium salts of chitosan combined with Fe_3_O_4_ nanoparticles can act as a bioadsorbent for methyl orange and chromium (VI) [31]. Similarly, a hydrogel produced by the cross-linkage of chitosan with glyoxal, glutaraldehyde, and terepathaldehyde is effective for organ transplant purposes and for the restoration of organ function [32]. Ho-166, Sm-153, and Lu-166 radionuclides cross-linked with chitosan have been used for targeted radiation therapy [33].

## 4. Synthesis of Chitosan Nanoparticles

Various procedures are used to synthesize CNPs due to chitosan’s ability to form a gel and to form beads [7]. The most prevalent methods used to prepare CNPs include ionic gelation, microemulsion, emulsion-based solvent evaporation, and emulsification solvent diffusion [34]. These processes generally involve non-complex procedures, utilizing less organic solvents during preparation. The main characteristics that have a direct bearing on the particle size, crystallinity, and surface charge of the CNPs obtained using these preparation techniques are molecular weight, the concentration of chitosan used, and the degree of deacetylation of the chitosan. A short synopsis of the procedures used to synthesize CNPs that are discussed in this review is shown in Table 1.

### 4.1. Ionic Gelation Method

Ionic gelation involves dissolving chitosan, which is positively charged, in an acetic acid solution at room temperature and magnetically stirring for an hour. A second solution is made using polyanion tripolyphosphate (TPP) dissolved in deionized water (DI). Generally, TPP is used as an ionic cross-linker [35]. The ionic gelation synthesis involves mixing an aqueous solution containing chitosan and another containing TPP, thus resulting in a complex coacervate aqueous phase [36]. The TPP–chitosan mixture needs to be magnetically stirred at room temperature [37]. The solution results in three individual phases depending upon the stage of the procedure, starting with clear (chitosan solution), followed by opalescent or milky (after adding TPP to the chitosan solution), and, finally, aggregated (after adding more TPP to a milky solution), whereby the milky appearance is the sign of the formation of CNPs. The harvested CNPs are highly suitable for drug delivery applications either in vitro or in vivo [35].

There are a number of examples of drug delivery applications using CNPs formed via the ionic gelation process. Gulati [38] used the ionic gelation process to form and evaluate the sumatriptan succinate-loaded CNPs, which were delivered through an intranasal system for migraine therapy. The goal of the study was to investigate the application’s therapeutic efficacy and whether this approach may lead to a reduction in dosing frequency. The formulation of thymoquinone (TQ)-encapsulated CNPs for Alzheimer’s disease using ionic gelation, targeting through nose-to-brain, was studied by Alam et al. [39]. During synthesis, the authors used a variable ratio of TQ to chitosan in the solution prior to the growth of the nanoparticles [39]. Alishahi et al. investigated CNPs loaded with vitamin C to test their effect on the immune system of trout [40]. The authors used ionic gelation with TPP to synthesize the CNPs, followed by ionotropic gelation to load the CNPs with vitamin C. The authors purport that their in vivo studies show that the vitamin C-encapsulated CNPs are effective in producing immune-induced activity in rainbow trout. Several researchers have used modified versions of the ionic gelation method to prepare drug delivery systems for various purposes. Saha et al. prepared chitosan nanoparticles loaded with ampicillin trihydrate via ionotropic gelation [41]. In this process, TPP was added to an aqueous solution containing the CNPs and ampicillin trihydrate via magnetic stirring at room temperature, resulting in the loading of the nanoparticles with the drug. The antimicrobial efficacy of the ampicillin trihydrate-loaded CNPs was tested on the *Staphylococcus aureus* strain and was determined to be considerably higher than that of CNPs alone. Trapani et al. [42] prepared dopamine-loaded CNPs via the modified ionic gelation method in order to test their potential for the treatment of Parkinson’s disease. In vivo exaperiments on a rat brain showed that the dopamine-loaded CNPs induce increased levels of striatal dopamine output that depend upon dosing.

### 4.2. Microemulsion Method

The microemulsion method used for the synthesis of CNPs has been reviewed by Yanat and Schroën [43]. In this method, the CNPs are synthesized using a suitable reverse micelle. The reverse micelle is formed upon introducing a surfactant into an organic solvent and then adding the mixture to an appropriate acidic solution containing chitosan. Banerjee et al. used the microemulsion method to prepare chitosan nanoparticles in a 1,4-bis-2-ethylhexylsulfosuccinate (AOT) and N-hexane reverse micelle mixed with chitosan in an acetic solution. [44]. Glutaraldehyde was added to the solution at room temperature, and the solution was then stirred overnight to accomplish the reaction with the free amine group of chitosan. The glutaraldehyde in this method acts as a cross-linker for chitosan [45]. Excess surfactant is removed by precipitation with CaCl_2_ and then centrifuged. The reverse miscellar method is well known for achieving a greater uniformity of size of nanoparticles. Banerjee et al. reported that the CNPs’ size varied from 30 to 110 nm with the size depending upon the degree of cross-linking of the chitosan that was accomplished using glutaraldehyde [44].

Hu et al. used 1-ethyl-3-(3-dimethylaminopropyl)carbodiimide (EDC) to promote a coupling reaction between stearic acid (SA) and chitosan oligosaccharide (CSO), resulting in a micelle-like nanoparticle formation between the two constituents [46]. The CSO-SA micelle-type nanoparticles were loaded with plasmid DNA in order to test its efficacy as a gene delivery system. Manchanda and Nimesh [47] used the same microemulsion method as Banerjee et al. [44] to prepare CNPs cross-linked with glutaraldehyde. The CNPs were loaded with synthetic oligonucleotides (ODNs) to study their release in vitro under varying pH conditions. The authors concluded that the release of ODNs was higher under basic conditions than at neutral or acidic conditions.

Brunel et al. [48] used the reverse micellar method to prepare CNPs and concluded that control over the particle size and size distribution is increased with a decrease in the MW value of chitosan. The authors attributed this to either a lowering of the viscosity of the aqueous droplets or the fact that chitosan polymer chains become increasingly disentangled as the MW is lowered. Fang et al. used the reverse micellar method, whereby a reverse phase suspension from the Span-80 emulsifier and glutaraldehyde was formulated, to prepare Fe_3_O_4_ magnetic CNPs [49]. In addition, CNPs loaded with bovine serum albumin (BSA) were prepared via the reverse micellar method by Kafshgari et al. [50]. From this work, the authors concluded that decreasing the chitosan concentration increased the release of BSA.

### 4.3. Emulsification Solvent Diffusion Method

In emulsification solvent diffusion, an emulsion is created by adding an organic phase to a chitosan-bearing solution that contains a stabilizing agent (e.g., poloxamer): the entire mixture needs to be mechanically stirred and then homogenized under pressure. Niwa et al. [51] used a modified emulsification method, which was first reported by El-Shabouri [52], in which they employed a D,L-lactide/glycolide copolymer (PLGA). In this method, polymeric precipitation and nanoparticle formation occur when the emulsion is diluted with liberal amounts of water. The diffusion of the organic solvent in the aqeous media is the governing mechanism of the emulsification solvent diffusion method. The disadvantages of this method include the substantial shear forces that occur during nanoparticle formation and organic solvent use.

### 4.4. Polyelectrolyte Complex Method

The polyelectrolyte complex (PEC) method involves the formation of CNPs by adding oppositely charged polymers or counter ions to chitosan in solution. The oppositely charged polymer or counter ion is typically dissolved with chitosan in an acetic acid solution while stirring under ambient conditions [53]. The PEC formulation occurs on account of the electrostatic interaction between the oppositely charged chitosan and the additional polymer or counter ion, resulting in charge neutralization. Thus, because of the charge neutralization, the PEC is self-assembled, leading to a substantial reduction in hydrophilicity [35]. The nanoparticles formed by this technique were reported to be from 50 to 700 nm in size [35]. These PEC-complexed CNPs have been used as delivery systems for drugs, proteins, peptides, and plasmid DNA. In addition, Liu and coworkers [54] used the PEC method to prepare CNPs loaded with heparin. They considered the effect of the pH of the solution, molecular weight (MW), and the concentration of the constituents on the yield size of nanoparticles. Their conclusions were that lower pH conditions and moderate MW values resulted in greater nanoparticle complexation. Sharma et al. [55] concluded that the IgA-loaded chitosan–dextran nanoparticles formed using the PEC method provide a simple and effective way to create a drug delivery system. The CNPs prepared by the authors had a size range of 300–500 nm with Zeta potential (see Section 5.7) values of +40 to +50 mV.

## 5. Characterization of Chitosan Nanoparticles

Characterization is critical for developing a full understanding of the formation mechanism, as well as the physicochemical and medicinal properties, of CNPs. The characterization of CNPs for the elucidation of their physicochemical properties has been considered in a number of studies. Herein, we provide a brief summary of the prevalent characterization techniques utilized for the interrogation of the physicochemical properties of CNPs.

### 5.1. Measurement of the Size of Chitosan Nanoparticles

The size of CNPs plays a critical role in regulating its various properties, such as conjugation power with drugs, crystallinity, and overall charge. The size determination of CNPs is generally made using either transmission electron microscopy (TEM) or dynamic light scattering (DLS). The use of DLS for size determination is often complicated because of the high polydispersity of CNPs. Characterization for information pertinent to the morphological, optical, and structural properties of CNPs can be obtained using various complementary techniques. This includes X-ray diffraction (XRD), Raman spectroscopy, TEM, scanning electron microscopy (SEM), and atomic force microscopy (AFM). UV-vis spectroscopy is occasionally used to determine concentrations of solutions containing CNPs, but will not be considered here.

### 5.2. X-ray Diffraction

The nature and degree of crystallinity of CNPs can be studied using X-ray diffraction (XRD). XRD is the foremost analytical tool used to identify the crystalline phase(s) of materials. In addition to determining structural properties, XRD can also be used to measure the mean diameter of the nanoparticles [56]. When measured in powder X-ray diffraction, the Scherrer equation [57] may be used to determine the mean crystallite size of the material after corrections in the broadening of the diffraction peaks due to instrumental, strain, and lattice imperfections. The Scherrer equation relates the full width with the half-maximum (FWHM) of a given XRD peak of a specific crystalline phase (after the aforementioned corrections) to the mean size of the nanoparticles, assuming the typical size of a nanoparticle is the same as the size of a crystallite [58]. The XRD patterns of chitosan nanoparticles synthesized using ionic gelation in the study by Ali et al. [59] are shown in Figure 2. XRD analysis revealed characteristic broad peaks occurring at 2θ ≈ 11° and 19.6° for chitosan, which are essentially extinguished upon the formation of the CNPs due to cross-linking with TPP [59]. The lack of distinct peaks in the XRD pattern measured from the CNPs is indicative of their amorphous structure, which is in agreement with a study carried out by Sivakami et al. [60]. Karavelidis et al. discovered that higher drug release rates were observed in nanoparticles produced from aliphatic polyesters that have a low degree of crystallinity, which were used to encapsulate ropinirole HCl [61]. Vaezifar et al. attributed the decrease in crystallinity of CNPs to the interpenetration of dense counter ions of TPP inside chitosan nanoparticles and the disruption of the complex network formed from long-chain polymers [62].

### 5.3. Raman Spectroscopy

Raman spectroscopy is a sensitive technique used to detect the structural modifications in macromolecular complexes, as deduced from the excitation of the associated Raman-active vibrational modes [63]. The Raman effect is based on the scattering of light, which includes both elastic scattering at the same wavelength as the incident light and inelastic scattering at a shifted wavelength, which is due to the excitation of a specific molecular vibration. In a study by Gordon et al. [64], Raman spectroscopy was used to test whether the adsorption of ovalbumin (OVA) to CNPs caused conformational changes to the protein. As shown in Figure 3, the Raman spectrum measured from OVA-loaded CNPs differs either from the spectra of the two individual components or from the spectrum of a simple physical mixture of the two components, demonstrating a conformational change in the OVA protein upon loading. The resultant hydrogen bonding, ion–ion, and ion–dipole interactions account for the conformational alteration of OVA. As demonstrated by Yamasaki et al., more subtle effects can be observed in the Raman spectra as a result of conformational changes, including frequency shifts, intensity changes, and line broadening of the pertinent Raman bands [65]. There is evidence for conformational modifications to vaccines having a direct bearing on the immune response of a host [66].

### 5.4. Transmission Electron Microscopy (TEM)

TEM can be used to study the degree of crystallinity, size, and morphology of nanoparticles. TEM uses a beam of focused high-energy electrons that transmits the specimen, whereby the resulting image can be viewed on an imaging system (i.e., phosphor screen) situated on the opposite side of the impinging electron beam. The image results from the interference between the electron beam that is transmitted through the sample and the beam that is diffracted from the sample. Typically, high resolution TEM imaging is possible to a length scale on the order of 2 Å. For TEM analysis, the sample needs to be very thin (~500 Å or less) to produce a high-resolution image [35]. TEM specimens can be prepared by using a negative staining material (e.g., uranyl acetate) or by the simple deposition of a dilute suspension of the sample on a carbon-coated copper grid. In a study by Ghadi et al. [67], TEM was used to analyze the magnetic chitosan nanoparticle’s shape and particle size. As shown in Figure 4, the Fe_3_O_4_ particles used in the study were coated well with chitosan and the diameter of CNPs ranged from 10 to 80 nm [67].

### 5.5. Atomic Force Microscopy (AFM)

The AFM can be used to image the nanoparticles with atomic resolution in a three-dimensional surface profile and to measure the force exerted by the sample surface on the AFM tip at the nano-Newton scale. Fauzi et al. [68] used the AFM to study the surface morphology of chitosan/maghemite nano-composite thin films with the aim of potentially applying the films for the optical detection of Hg^2+^ ions via surface plasmon resonance. Figure 5 shows the AFM three-dimensional surface profile of chitosan, maghemite, and chitosan/maghemite nano-composite thin films. The surface morphology can be quantified using the root-mean-square (RMS) roughness value [68], which is calculated either using the cross-sectional or a two-dimensional profile [69]. The RMS roughness of the chitosan, γ-Fe_2_O_3_, and chitosan/γ-Fe_2_O_3_ was found to be 1.4 nm, 47 nm, and 37.3 nm, respectively. The authors concluded that a smoothening of the meghamite surface was induced by the deposition of chitosan, whereby the smoothening mechanism was attributed to surface diffusion [70]. The resulting surface morphology introduced in the nanostructured chitosan/maghemite composite thin film was deemed to be appropriate to enhance the sensing of Hg^2+^ [70]. Almalik et al. [71] used AFM to study chitosan nanoparticles that were coated with hyaluronic acid (HA). In this study, AFM was used to determine the presence of an HA on CNPs. The thickness of the HA coverage on the CNPs was estimated to be approximately 20–30 nm when dry.

### 5.6. Scanning Electron Microscopy (SEM)

SEM utilizes a focused high-energy beam of electrons to image the surface of a specimen. Other than in an environmental SEM that will not be discussed herein, samples are analyzed upon drying. Non-conducting samples need to be sputter-coated with a thin layer of carbon or metal. SEM provides a direct image of the surface morphology and size of nanoparticles. For SEM’s equipped with an element dispersive spectroscopy (EDS) detection capability, the elemental composition of samples can be studied from the collected emission X-ray spectra. Saharan et al. [72] prepared chitosan nanoparticles, chitosan–saponin nanoparticles, and Cu-CNPs using ionic gelation to test their effect individually as antifungal agents. In their study, SEM was used to confirm the organization of their chitosan nanoparticles on the nanometer scale. In addition, SEM imaging showed the chitosan–saponin nanoparticles to be spherical in shape, whereas Cu-CNPs had the shape of a compact polyhedron [72]. Jingou et al. [73] used SEM for a morphological study of CNPs that were cross-linked with TPP and loaded with a combination of salicylic acid and gentamicin. Their investigations showed that these CNPs were nearly spherical in shape with an average size of ~200 nm [73].

### 5.7. Dynamic Light Scattering (DLS)

Through detection of the scattered light intensity, the DLS technique enables the measurement of the dynamic fluctuations of particles stemming from their Brownian motion in a solution. The light source is monochromatic and is typically from a laser, and the particles in the solution produce a dynamic diffraction of light in a speckel pattern. The average particle size within a sample can be determined using the Stokes–Einstein equation. In addition, DLS enables the determination of the polydispersity of nanoparticles within a sample (i.e., the polydispersity index (PDI)), or the dispersity in the new convention as recommended by the IUPAC, and their Zeta potential. The use of the DLS technique for nanoparticle size determination typically requires monitoring intensity variations at various detector angles for several polarizations relative to the incident light [74,75]. In a study by Fan et al. [76], DLS was performed on CNPs, saponin-CNPs, and Cu-CNPs at a light-scattering angle of 90°. The size distribution profiles are shown in Figure 6a–c: the authors report that, based on these measurements, the mean diameter and the PDI or dispersity values of CNPs, saponin-loaded CNPs, and Cu-loaded CNPs were ~192 nm and 0.6, ~374 nm and 1.0, and ~196 nm and 0.5, respectively [72]. The lower PDI values show that the CNPs and Cu-CNP nanoparticles have narrower size distributions than the saponin-CNPs. The authors reported that the CNP and Cu-CNP samples had larger Zeta potential values, of +45.33 and +88 mV, respectively, compared to the value reported for saponin-CNPs (+31 mV) [73]. The Zeta potential, which is a measure of the electric potential between the attached and non-attached fluid with respect to the nanoparticle surface, is a measure of the stability of nanoparticles in the fluid. Therefore, the CNP and Cu-CNPs, having larger Zeta potential values, exhibit higher stability than saponin-CNPs in an aqueous media [77]. In a study by Lu et al. [78], CNPs loaded with DNA and grafted with polyethylenimine (PEI) were prepared for gene therapy for osteoarthritis. Using DLS to measure the dispersity and Zeta potential, the authors reported that particle size decreased, whereas surface charge increased (i.e., larger Zeta potential), as the CNP:DNA weight-to-weight ratio increased.

## 6. Nanomedicinal Application of Chitosan Nanoparticles

Based on current trends, it is conceivable that nanomedicine will help to bring the next leap in developing advanced therapy, imaging, drug delivery, and the treatment of fatal diseases. Chitosan nanoparticles are natural materials that are widely used in medicinal applications due to their hydrophilic, nontoxic, biocompatible, and bio-degradable nature. Because CNPs have these properties, they are highly suitable for a wide range of drug delivery, gene therapy, and tissue engineering applications. Some of these applications are discussed below.

### 6.1. Drug Delivery

The demonstrated potential use of CNPs as drug delivery systems has provided opportunities for the development of a largely expanded range of CNP-based delivery vehicles [79]. Due to its biocompatibility, chitosan is classified by the United States Food and Drug Administration as GRAS (Generally Recognized as Safe) [80]. As noted above, the presence of the amino and hydroxyl functional groups, as well as the glycosidic bond, enables the loading of CNPs with drug molecules and DNA. Because CNPs are soluble in acidic aqeous solutions, sustainable chemistry may be employed in their synthesis without the use of harmful organic solvents [81]. An additional advantage of using CNPs is that, through mucoadhesion, they enable the controlled release of drugs in vivo [82]. CNPs have several important potential applications for the delivery of drugs, namely, parenteral, ocular, oral, pulmonary, nasal, buccal, and vaginal, in addition to applications in cancer therapy, tissue engineering, etc. [34]. A few of these applications of CNPs for drug delivery are discussed below.

#### 6.1.1. Ocular Drug Delivery

Because chitosan has mucoadhesive properties, the use of CNPs for controlled drug delivery is favorable via mucosal membranes [83]. CNPs undergo surface gel layer formation when in contact with near-neutral aqueous fluids, which may improve residence time on the mucosal surface and the efficacy of drug delivery to ocular tissue [84]. CNPs cross-linked using sulfobutylether-β-cyclodextrin (SBE-β-CD) were utilized for investigations of their potential for ocular drug delivery by Mahmoud et al. [85]. These authors used econazole nitrate (ECO) to test for ocular antifungal efficacy in albino rabbits. Their results showed that the prepared CNPs were predominantly pseudospherically shaped with average particle sizes ranging from 90 to 673 nm and Zeta potential values ranging from 22 to 33 mV. The ECO drug loading percent values ranged from 13 to 45% [85]. The authors performed in vivo studies, which showed that the ECO-loaded CNPs had better antifungal ocular efficacy than an ECO solution, thus confirming that chitosan nanoparticles are a promising ECO drug delivery vehicle for antifungal ocular treatment [86]. Santhi et al. [86] used the emulsification preparation technique to synthesize fluconazole-loaded CNPs with an average particle size of 152.85 ± 13.7 nm. The authors used the cup-plate method to test the efficacy of antifungal treatment using fluconazole-loaded CNPs as compared to that of fluconazole eye drops [86]. The fluconazole-loading capacity of their CNPs was found to be optimal at ≤ 50%. The authors concluded that the CNPs exhibited promising characteristics, including drug loading capacity, antifungal activity, and prolonged drug release, for fluconazole drug delivery for antifungal treatment [86].

#### 6.1.2. Oral Drug Delivery

Oral drug delivery is widely used because of several factors, including convenient drug administration, controlled delivery, low production cost, and patient compliance: However, challenges in conventional oral drug delivery include drug solubility issues in low-pH gastric fluids, the degradation and reduced activity of drugs due to the presence of enzymes, and the lack of adequate membrane permeability [3]. Nanomedicine offers potential opportunities to overcome such challenges in oral drug delivery [87,88]. The various physicochemical properties of CNPs noted above, including mucoadhesion, biocompatibility, large surface-to-volume ratios, and drug conjugation versatility, make them suitable candidates for improving oral drug delivery. Pan et al. [36] conjugated insulin to CNPs, which ranged 250–400 nm in size and were positively charged, and performed oral drug administration to diabetic rats. The authors reported that the CNP-assisted oral drug delivery resulted in the enhanced intestinal absorption of insulin in diabetic rats. By modulating the dose of insulin loaded on the CNPs, the authors found that glucose levels in diabetic rats could be brought to normal levels for an extended period of time. Although the precise mechanism is not known, the authors conjectured that the CNPs improve the stability of insulin by providing protection in the gastrointestinal environment, thereby helping to increase the drug uptake [36]. In order to study the potential improvement in the bioavailability of lipophilic drugs, such as cyclosporine, by nanoparticle encapsulation, El-Shabouri [49] used cyclosporine-A-loaded CNPs for oral administration in dogs, whose blood samples were analyzed at predetermined intervals after administration for drug uptake. The mean size of the chitosan HCl nanoparticles was 148 nm with a Zeta potential of +31 mV. The results from this study showed that cyclosporine A bioavailability was increased by 73% when administered via CNP encapsulation compared to oral delivery using the commercial microemulsion Neoral^®^ [49]. El-Shabouri conjectured that the positively charged CNPs interact more strongly with negatively charged epithelial cells of the gastrointestinal tract than neutral or negatively charged carriers, thereby resulting in greater permeability and bioavailability of the drug [49].

#### 6.1.3. Pulmonary Drug Delivery

There are several physiological properties of the lungs that may enable enhanced drug delivery, including a relatively thin absorption barrier for drug uptake, a large surface area, and their extensive vascularity [89]. As with oral drug delivery, pulmonary drug delivery is conjectured to benefit from the physicochemical properties of CNPs, including a large loading capacity, mucoadhesion, a positive charge, an antibacterial property, and sustained drug delivery. Islam and Ferro [90] have made an extensive review of CNP-based vehicles for pulmonary drug delivery. Several noteworthy studies have been made to investigate the efficacy of using CNPs for pulmonary drug delivery. Yamamoto et al. used a poly(DL-lactide-co-glycolide) (PLGA) copolymer for the synthesis of CNPs that were loaded with elcatonin (used for lowering blood calcium) and aerolized for pulmonary drug delivery [91]. The results from their studies demonstrated the efficacy of the drug delivery, along with sustained drug release (up to 24 h), using CNP-PLGAs loaded with elcatonin when compared to the use of unmodified CNPs. The authors conjectured that the positively charged properties of the PLGA-modified CNPs enabled the opening of the tight junctions in the epithelial cells of the lungs, thus improving drug uptake. Jafarinejad et al. [92] prepared aerosolized CNPs loaded with itraconazole, which is an antifungal drug, with in vitro testing for pulmonary administration. The study was made to test whether the low solubility of itraconazole in the gastrointestinal tract upon oral administration can be overcome by pulmonary drug delivery using CNPs. The authors reported an increased uptake of itraconazole using their aerosolized CNPs, particularly when leucine was added for aerosolization of the nanoparticles [92]. Rawal et al. prepared CNPs loaded with rifampicin for pulmonary administration in rats in order to test for the efficacy of drug delivery in the treatment of tuberculosis [93]. The primary advantage in pulmonary drug delivery is the potential of eliminating the considerable adverse effects to the drug when administered orally. The in vitro study showed sustained release of the drug for up to 24 h and negligible toxicity.

#### 6.1.4. Nasal Drug Delivery

The effective administration of peptides, nucleic acids, vaccines, and other drugs encapsulated in nanoparticles via nasal delivery is highly desirable because this route may induce a substantially more potent immune response; however, the nasal epithelium presents low permeability to hydrophilic molecules, whereas mucosal clearance and the mucus gel inhibit drug uptake in nasal passageways [81]. Due to their mucoadhesion, biocompatibility, low toxicity, and other properties, chitosan nanoparticles are postulated to be good candidates for an effective nasal delivery of drugs [94]. Shahnaz et al. synthesized thiolated CNPs (chitosan conjugated with thioglycolic acid) loaded with leuprolide, which is used to treat prostate cancer, the lining of the uterus, and uterine fibroids, to test whether this formulation could improve the bioavailability of the drug via nasal delivery [95]. Their results showed substantially improved bioavailability of leuprolide from nasal administration using thiolated CNPs in rats compared to the administration of leuprolide solution alone. One of the main complications in the conventional oral delivery of anti-epileptic drugs is the prevention of drug uptake in the brain due to the blood–brain barrier, resulting in drug resistance. Liu and coworkers [96] used carboxymethyl-CNPs loaded with carbamazepine, which is an anti-epileptic drug, to study the bioavailability of the drug when administered intranasally. The authors used CNPs that were found to have a particle size of ~219 nm with high entrapment efficiency (80%) [96]. From in vivo tests in mice, the authors concluded that carboxymethyl-CNPs carriers caused substantially improved bioavailability and enhanced brain-targeting of carbamazepine, when compared to the nasal administration of a carbamazepine solution [96]. However, the authors report that the CNP-carrier administration is limited by the volume of the drug that can be delivered intranasally.

#### 6.1.5. Buccal Drug Delivery

Buccal drug delivery is a preferred route for the delivery of drugs, particulary ones with high molecular weights, that cannot be administrated by the oral route [97]. This is a transmucosal delivery mechanism that has advantages over the oral route, including overcoming drug degradation in the gastrointestinal tract and the first-pass metabolism effect. Thus, the buccal drug delivery method may lead to enhanced drug bioavailability and lower required doses of the drug [98,99]. Mazzarino et al. [100] prepared films containing CNPs coated with polycaprolactone and loaded with curcumin, which has potential uses for the treatment of periodontal disease, for administration via the buccal mucosa route. AFM and SEM characterization of the films showed the confirmed presence of CNPs in the films and that they were uniformly distributed throughout the films. In vitro studies conducted by the authors in simulated saliva solutions showed maximum swelling of the films due to a hydration of ~80% and the sustained delivery of curcumin, which are required for the successful treatment of periodontal disease.

#### 6.1.6. Vaginal Drug Delivery

As with buccal drug delivery, the vaginal mucosa offers another transmucosal route for drug administration. Drugs are administered vaginally for two approaches: either for local treatment or for systemic effects whereby the drug passes through the vaginal mucosa and enters the bloodstream [101,102,103]. Primarily due to their mucoadhesive and conjugation properties, chitosan nanoparticles may be ideally suited as drug carriers in vaginal drug administration for systemic effects. Nevertheless, there are several challenges for vaginal drug delivery including a low pH (3.8–4.5) vaginal environment, considerable fluid discharge, and extensive epithelial tissue folding. Martínez-Pérez et al. [104] prepared CNPs that were surface modified with PLGA and loaded with clotrimazole for vaginal drug administration. In vitro studies revealed that delivery via CNP-PLGAs improved antifungal activity in relation to delivery without the use of CNPs, thus making CNP-PLGAs potentially useful for the treatment of fungal infections of the vagina [104]. Similarly, in a separate study, Perineli et al. [105] developed a hydrogel system containing hydroxypropyl methylcellulose (HPMC) and chitosan to treat fungal infection of the vagina caused by *Candida albicans* and non-albicans strains. The HPMC-chitosan hydrogel was found to possess either CNPs or monomolecular chitosan and was loaded with metronidazole. In vitro studies and mucoadhesive tests revealed that both types of HPMC-chitosan hydrogel, whether containing CNPs or monomolecular chitosan, exhibited improved anti-*Candida* activity of all strains and enhanced mucoadhesive properties [105].

### 6.2. Cancer Therapy

Chemotherapy remains an important therapy route for the treatment of cancers; however, because cytotoxic chemotherapeutic drugs cause chemical damage to both cancerous and noncancerous cells, this option produces considerable adverse effects in patients. Chitosan-based nanostructures (i.e., nanoparticles, nanocomposites, nanorods, etc.) are one class of polymer-based nanomaterials that are projected to play an important role in providing cancer-targeted therapies under controlled drug release that will minimize adverse effects. Mathew et al. [106] prepared CNPs decorated with Mn-doped ZnS quantum dots as drug carriers and as a cancer cell imaging agent using fluorescent microscopy. In vitro studies using the CNPs loaded with 5-Fluorouracil on the MCF-7 breast cancer cells revealed that these carriers are useful for controlled and targeted drug delivery. Sekar et al. [107] synthesized CNPs loaded with ascorbic acid for tests of efficacy in potential targeted drug delivery in the treatment of cervical cancer. The authors found from in vitro studies that the ascorbate-CNPs are effective in targeting HeLa cervical cells with no effect on human-diploid fibroblast (WI-38 strain) normal cells, thus demonstrating the potential use of these carriers as cancer drug delivery systems. Nascimento et al. [108] developed chitosan-polyethylene glycol (PEG) nanoparticles loaded with silencing RNA (siRNA) that were specifically designed to target epidermal growth factor receptor (EGFR) proteins and silence the overexpressing Mad2 gene in tumor cells. The authors report that the use of CNP-PEG-siRNA nanoparticles loaded with cisplatin, in comparison to CNP-PEG-siRNA alone, had dramatically stronger inhibiting effects against cisplatin-resistant tumors in the lung. In addition, the CNP-PEG-siRNA-cisplatin nanoparticles enabled considerably reduced drug dosage with negligible adverse effects. Another approach to targeting tumor cells involves using glycol chitosan, which, due to its surfactant properties, can be used for the self-assembly of glycol-CNPs encapsulating tumor-targeting drugs [109]. Hydrophobic camptothecin (CPT) (a topoisomerase I inhibitor) was encapsulated in glycol-CNP drug carriers by Min et al. [110] for studies of anticancer efficacy. Tests on MDA-MB231 human breast cancer xenograft models indicated that the CPT-loaded glycol-CNPs had much higher antitumor efficacy than glycol-CNPs alone.

### 6.3. Tissue Engineering

Due to their biocompatibility, biodegradability, non-toxicity, antibacterial activity, functionalizability, and other properties mentioned above, chitosan-based nanostructures are increasingly being used in and investigated for tissue engineering applications [111,112,113,114]. Chitosan-based nanostructures have found applications in bone, periodontal, blood vessel, skin, corneal, and cartilage tissue engineering [111]. Chitosan is typically combined with other biopolymers or with bioactive nanoscale ceramic materials to formulate scaffolding for use in tissue engineering. One of the more active areas in the application of chitosan-based scaffolds is for periodontal tissue engineering [111,114]. Studies made on chitosan-based scaffolds generally show good efficacy in periodontal tissue engineering, but have poor mechanical strength. Investigations in utilizing bioactive bioceramics (e.g., hydroxyapatite, Bioglass, etc.) in composite form with chitosan in scaffolding may lead to improved mechanical strength properties [112]. Another noteworthy area of application of chitosan-based scaffolds is for corneal tissue engineering. Tayebi et al. [115] recently developed a transparent composite scaffolding containing CNPs and polycaprolactone (PCL). Culture studies showed that human corneal endothelial cells attached appropriately to the CNP-PCL scaffolding in monolayer formation, indicating significant potential for corneal tissue engineering applications [115].

## 7. Conclusions

In this review paper, we have made a summary of the types of synthesis and characterization techniques used in the study of chitosan nanoparticles, along with their various nanomedicinal applications. As one of the most abundant polysaccharides with all of the essential characteristics, such as biodegradability, non-toxicity, biocompatibility, hydrophilicity, antibacterial ability, etc., CNPs are one of the best options for nanomedicinal application. CNPs show excellent potential as carriers for the encapsulation and incorporation of drug molecules. Thus, in association with their mucoadhesive properties, CNPs show good potential for effective drug delivery, controlled drug release, and the enhanced therapeutic efficacy of the drugs. The fact that CNPs demonstrate favorable in situ gel formation ability and mucoadhesive characteristics, and are positively charged, makes them effective for pulmonary and ocular drug delivery. Similarly, their ability to open tight junctions of the mucosal membrane and enhance drug absorption properties makes CNPs promising oral and pulmonary drug delivery carriers, whereas the ability to increase the permeability of various drugs makes them suitable for nasal drug delivery. Investigations made on CNP-based nanostructures loaded with appropriate anticancer drugs show that these can be highly effective in targeted cancer therapy at lower dosing levels that result in considerably less adverse effects than in conventional chemotherapy. Chitosan-based nanostructures in composite formulation with other polymers and bioactive bioceramics for scaffolding show considerable promise for applications in tissue engineering. However, further investigations of the potential toxicity of CNPs, dependent upon the precise nature of anticipated application, need to be performed to ensure complete safety prior to industrial application. Our hope is that more CNP-based applications can be developed for use in treatment, therapy, imaging, and drug delivery to cure cancers and other fatal diseases.

## Figures and Tables

**Figure 1 nanomaterials-13-01302-f001:**
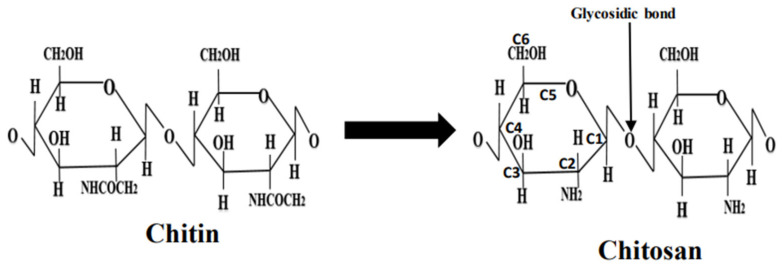
Molecular conversion of chitin to chitosan.

**Figure 2 nanomaterials-13-01302-f002:**
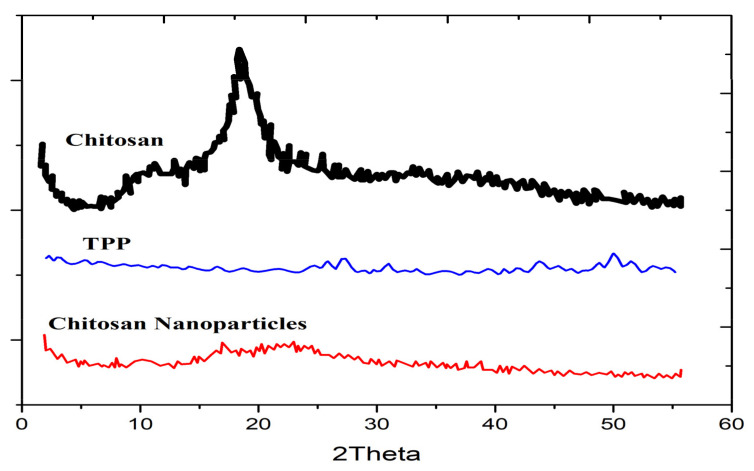
XRD patterns of pure chitosan, TPP, and chitosan nanoparticles reproduced from Ali et al. [59].

**Figure 3 nanomaterials-13-01302-f003:**
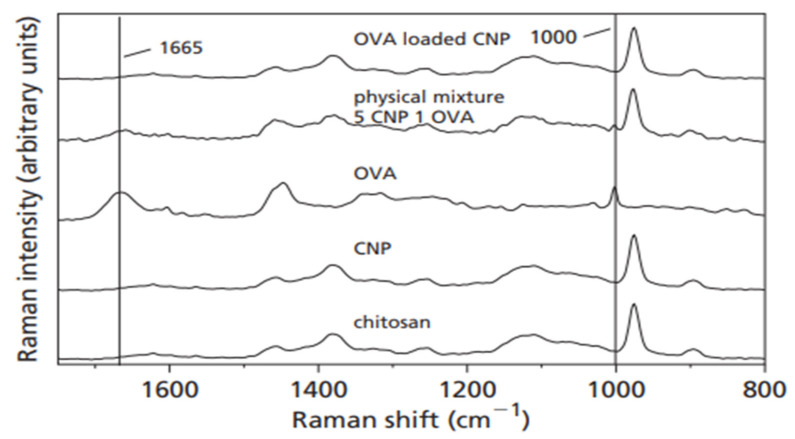
Raman spectra of chitosan, chitosan nanoparticles (CNPs), OVA, a physical mixture of 5:1 ratio of CNPs to OVA, and OVA-loaded CNPs (5:1 ratio of OVA to CNPs) [64].

**Figure 4 nanomaterials-13-01302-f004:**
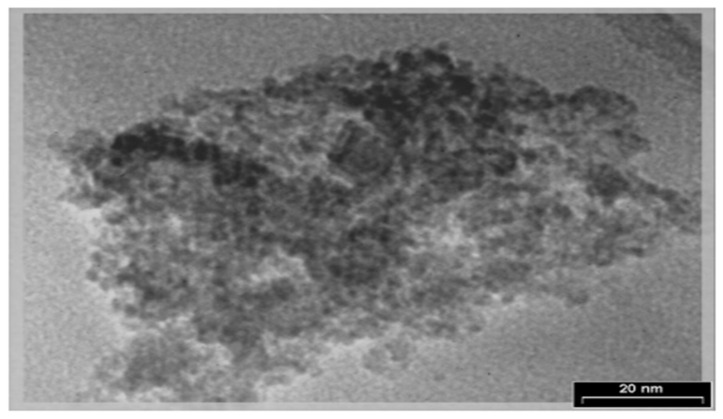
A TEM image of chitosan nanoparticles [67].

**Figure 5 nanomaterials-13-01302-f005:**
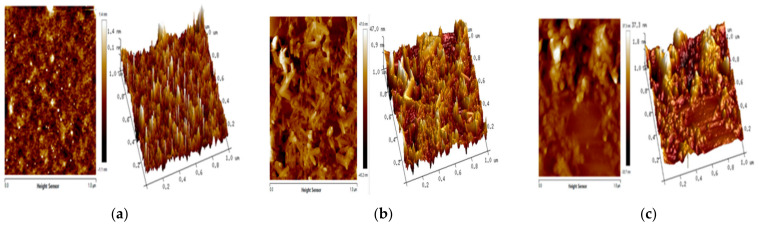
Atomic force microscopy image of (**a**) chitosan thin film, (**b**) Fe_2_O_3_ thin film, (**c**) and a composite chitosan/Fe_2_O_3_ thin film [68].

**Figure 6 nanomaterials-13-01302-f006:**
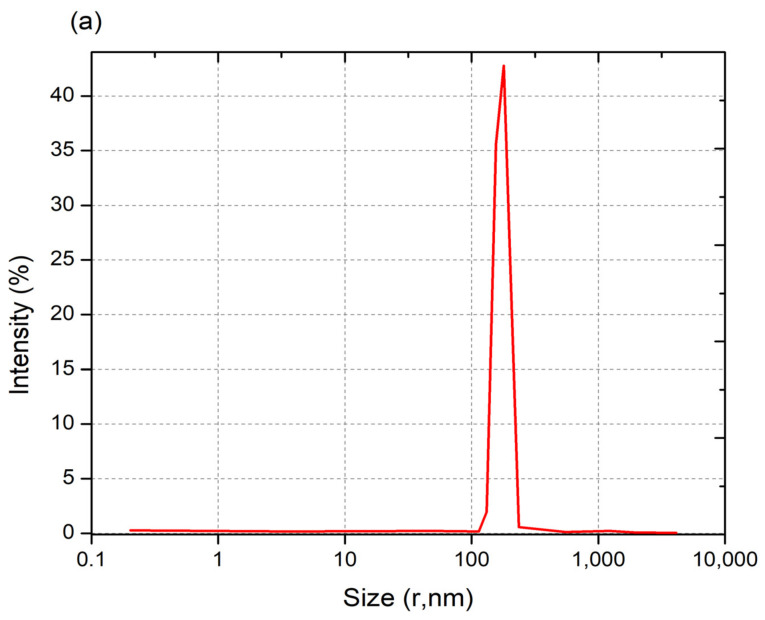
The size distributions as measured using DLS of (**a**) CNPs, (**b**) saponin-CNPs, and (**c**) Cu-CNP nanoparticles reproduced from Saharan et al. [72].

**Table 1 nanomaterials-13-01302-t001:** Synthesis methods used to make chitosan nanoparticles.

Methods	Process	Advantages	Disadvantages
Ionic gelation	Ionic cross-linking activated by mixing an aqueous solution containing chitosan and another containing TPP, thus resulting in a complex coacervate aqueous phase.	Straightforward procedure using mild chemicals. NP size easily regulated by altering the concentration of chitosan and TPP.	Difficult to produce uniformly sized NPs.
Microemulsion/reverse micelles	Based on covalent cross-linking where reverse micelle is formed upon introducing a surfactant into an organic solvent and then adding the mixture to an appropriate acidic solution containing chitosan.	Straightforward procedure achieving greater uniformity of size of NPs.	Use of harmful chemicals and a time intensive process.
Emulsification solvent diffusion method	Polymeric precipitation resulting in the formation of nanoparticles.	Straightforward procedure.	Substantial shear forces occur during the formation of CNPs.
Polyelectrol yte complex method	A self-assembly occuring due to the electrostatic interaction between the oppositely charged chitosan and the added polymer or counter ion, resulting in charge neutralization.	NP size can be regulated by pH of the solution, molecular weight (MW), and concentration of the constituents.	Due to the neutralization of charge, the PEC is self-assembled, leading to a substantial reduction in hydrophilicity.

## Data Availability

Not applicable.

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
