# Peer review of "A Review of the Preparation, Characterization, and Applications of Chitosan Nanoparticles in Nanomedicine"

_nanomaterials, 2023, doi:10.3390/nano13081302_

Round 1

Reviewer 1 Report

This presented review paper attempted to present the modern advances in chitosan chemistry, focusing not only on the synthesis and characterization methods but also on the potential applications in drug delivery/cancer therapy, which covered many areas. Although this field of chemistry is very broad, the authors have succeeded in presenting new results in this area and this study deserves to be published. However, three critical points need to be mentioned detailed as follows:

1, The quality of the figures 2. and 6. is really poor. You should improve their quality.

2, Five different synthesis methods are presented in the manuscript, but it is not clear whether there are advantages to using one or the another apart from the different particle size range that can be guaranteed. There is no information on how to choose the right synthesis method depending on the further application.

3, What are the advantages of using chitosan in comparison to other carriers, e.g. layered double hydroxides (LDH)? In order to present this properly, a comparative table need to be drawn up in which the most relevant data are inserted. For this purpose, among others, you should use the following article:

M. Pavlovic et al. “Surface modification of two-dimensional layered double hydroxide nanoparticles with biopolymers for biomedical applications” Adv. Drug Deliv. Rev. 191 (2022) 114590. DOI: https://doi.org/10.1016/j.addr.2022.114590

Considering my above mentioned criticisms, I recommend this review for publication in an MDPI journal “Nanomaterials” after minor revision.

Author Response

Journal: Nanomaterials-MDPI

Manuscript ID: nanomaterials-2272226

Title: "A Review of the Preparation, Characterization, and Applications of Chitosan Nanoparticles in Nanomedicine”

The change are highlighted.

Reviewer: 1

Comments:

This presented review paper attempted to present the modern advances in chitosan chemistry, focusing not only on the synthesis and characterization methods but also on the potential applications in drug delivery/cancer therapy, which covered many areas. Although this field of chemistry is very broad, the authors have succeeded in presenting new results in this area and this study deserves to be published. However, three critical points need to be mentioned detailed as follows:

Point 1: The quality of the figures 2. and 6. is really poor. You should improve their quality

We agree that the quality of Figures 2 and 6 is rather poor, and accordingly, we have generated our own figures showing reproduced data measured by the authors attributed in the figure captions, that are considerably better quality than the original figures in the references.

Point 2:  Five different synthesis methods are presented in the manuscript, but it is not clear whether there are advantages to using one or the another apart from the different particle size range that can be guaranteed. There is no information on how to choose the right synthesis method depending on the further application.

We agree with the reviewer that more information may be useful to the reader on use of a suitable synthesis method depending on the further application. We have added a table, with a table caption “Synthesis methods used to make chitosan nanoparticles”, where we summarize each of the synthesis methods and the advantages and disadvantages of each.

Point 3: What are the advantages of using chitosan in comparison to other carriers, e.g. layered double hydroxides (LDH)? In order to present this properly, a comparative table need to be drawn up in which the most relevant data are inserted. For this purpose, among others, you should use the following article.

We would like to thank the reviewer for the suggestion of comparison of chitosan with LDH. Indeed, the use of LDH for medicinal drug delivery offers a very promising carrier option. However, expanding our review to include use of LDH and other carriers is beyond the focus of our work: Our review is intended to be more narrow in scope and is strictly focused on the synthesis and applications of chitosan nanoparticles for drug delivery, cancer therapy and tissue engineering. There are a number of good reviews in the literature that cover a wide range of different types of nanostructured carriers used for drug delivery.

Reviewer 2 Report

The manuscript is a review article about the preparation, characterization and biomedical applications of chitosan nanoparticles. The paper is well organized and written. I beliew it provides a quite updated survey on chitosan nanoparticles. Hence, the paper can be accepted for publication in Nanomaterials after the following (minor) points are addressed:

1) Page 4, line 173. The meaning of the sentence is not clear. Please, rephrase it. 

2) Page 4,  line 181. "Efficacy" instead of "Effecacy"

3) Page 7, line 293. TEM and the acronym is repeated. 

4) Table 1 could be removed. The information reported here is well know, there is no need to report.

5) The quality (resolution) of the figures is very poor. Some of them appear  distorted. Please, improve (at least in the final version) the quality of the figures.

Author Response

Reviewer: 2

Comments:

The manuscript is a review article about the preparation, characterization, and biomedical applications of chitosan nanoparticles. The paper is well organized and written. I believe it provides a quite updated survey on chitosan nanoparticles. Hence, the paper can be accepted for publication in Nanomaterials after the following (minor) points are addressed:

Point 1: The meaning of the sentence is not clear. Please, rephrase it.

The sentence was rephrased as follows: “The solution results in three individual phases depending upon stage of procedure, starting with clear (chitosan solution), opalescent or milky (after adding TPP to the chitosan solution), and aggregated (after adding more TPP to a milky solution), whereby the milky appearance is the sign of the formation of CNPs.”

Point 2: line 181. "Efficacy" instead of "Effecacy".

Effecacy was changed to Efficacy.

Point 3: Line 293, TEM and the acronym is repeated.

The repeated acronym for TEM in line 293 was deleted as suggested.

Point 4: Table 1 could be removed. The information reported here is well know, there is no need to report.

We agree with the reviewer and have deleted the table.

Point 5: The quality (resolution) of the figures is very poor. Some of them appear distorted. Please, improve (at least in the final version) the quality of the figures.

We have reproduced the data to improve Figures 2 and 6 (see Point 1 for Reviewer 1). We deem Figure 3 (from reference 64) to be of sufficiently good quality. However, Figures 4, 5, and 6 are as presented in the original papers and we have no means to revised or reproduce these figures.

Reviewer 3 Report

The authors submitted a manuscript entitled “A Critical Review of the Synthesis, Characterization, and Nano-medicinal Applications of Chitosan Nanoparticles” with the reference nanomaterials-2272226.

The subject addressed by the authors is interesting and clearly deserves a review. The manuscript is globally well written. It is quite pedagogical and a good introduction to the subject.

However to be published the authors need to deep review the manuscript to solve some bugs and errors:

Line 80 – Figure 1 caption should be below figure and both should be prior to point 2 caption.

Line 201 – Correct “dependends”

Line 302 – Correct “ … can also be used …”

Line 450 – Correct “applations”

Line 597 – Correct “ … in a separate study by Perineli et al. [107] developed …”

Line 620 – Correct “Nasciamento” to “Nasacimiento”

More relevant the title in the pdf file “A Review of the Preparation, Characterization, and Applications of Chitosan Nanoparticles in Nanomedicine” does not fit with the one in the platform “A Critical Review of the Synthesis, Characterization, and Nano-medicinal Applications of Chitosan Nanoparticles”. Must be corrected.

When explaining the various methods the 4.5 – Reverse micellar method looks quite similar to 4.2 -Microemulsion method. Both uses reverse micelles induced by surfactants in organic or aprotic solvents. Could these two methods be considered together as one? If the authors want to keep these as separated methods, they should explain clearly which are the differences between them.

Author Response

Reviewer: 3

Comments:

The authors submitted a manuscript entitled “A Critical Review of the Synthesis, Characterization, and Nano-medicinal Applications of Chitosan Nanoparticles” with the reference nanomaterials-2272226.

The subject addressed by the authors is interesting and clearly deserves a review. The manuscript is globally well written. It is quite pedagogical and a good introduction to the subject.

However, to be published the authors need to deep review the manuscript to solve some bugs and errors:

Point 1: Figure 1 caption should be below figure and both should be prior to point 2 caption.

We have deleted the extra space so that the caption is immediately below Figure 1, as suggested, in the revised manuscript.

Point 2: Line 201 – Correct “dependends”

“Dependends” has been corrected to read as “depends”, as suggested.

Point 3: Line 302 – Correct “ … can also be used …”

We have corrected the sentence accordingly.

Point 4: Line 450 – Correct “applations”

We have changed “applations” to “applications” in the revised manuscript.

Point 5: Line 597 – Correct “ … in a separate study by Perineli et al. [107] developed …”

Perinelli was corrected to read as Perineli in the revised manuscript, on this line.

Point 5: Line 620 – Correct “Nasciamento” to “Nasacimiento”

“Nasciamento” was changed to “Nasacimiento” on line 620 in the revised manuscript.

Point 6: More relevant the title in the pdf file “A Review of the Preparation, Characterization, and Applications of Chitosan Nanoparticles in Nanomedicine” does not fit with the one in the platform “A Critical Review of the Synthesis, Characterization, and Nano-medicinal Applications of Chitosan Nanoparticles”. Must be corrected.

The title was changed to “A Review of the Preparation, Characterization, and application of Chitosan Nanoparticles in Nanomedicine” instead of “A Critical Review of the Synthesis, Characterization, and Nano-medicinal Applications of Chitosan Nanoparticles”.

Point 7: When explaining the various methods, the 4.5 – Reverse micellar method looks quite similar to 4.2 -Microemulsion method. Both uses reverse micelles induced by surfactants in organic or aprotic solvents. Could these two methods be considered together as one? If the authors want to keep these as separated methods, they should explain clearly which are the differences between them.

We agree with the suggestion from the reviewer. The two methods, i.e., reverse micellar and microemulsion, are quite similar and have been merged in the revised manuscript into the Microemulsion Method.
